# Health complexity assessment in primary care: A validity and feasibility study of the INTERMED tool

Camila Almeida de Oliveira[1]*, Bernardete Weber[2], Jair Lício Ferreira dos Santos[3], Miriane Lucindo Zucoloto[1], Lisa Laredo de Camargo[4], Ana Carolina Guidorizzi Zanetti[5], Magdalena Rzewuska[6,7], João Mazzoncini de Azevedo-Marques[3]

1 Public Health Postgraduate Program, Ribeirão Preto Medical School, University of São Paulo, São Paulo, Brazil, 2 Registered Nurse, Hospital do Coração (HCor), São Paulo City, Brazil, 3 Department of Social Medicine, Ribeirão Preto School of Medicine, University of São Paulo, São Paulo, Brazil, 4 Postgraduate Program in Psychiatric Nursing, Ribeirão Preto College of Nursing, University of São Paulo, São Paulo, Brazil, 5 Department of Psychiatric Nursing and Human Sciences, University of São Paulo at Ribeirão Preto College of Nursing, WHO Collaborating Centre for Nursing Research Development, São Paulo, Brazil, 6 Health Services Research Unit, University of Aberdeen, Aberdeen, Scotland, United Kingdom, 7 Aberdeen Centre for Health Data Sciences, University of Aberdeen, Aberdeen, Scotland, United Kingdom

☯ These authors contributed equally to this work.
* ocamila35@usp.br

**Data Availability Statement:** All relevant data files are available from the USP – Universidade de São Paulo database (http://repositorio.uspdigital.usp. br/handle/item/263).

## Abstract

### Background

Health complexity includes biological, psychological, social, and health systems. Having complex health needs is associated with poorer clinical outcomes and higher healthcare costs. Care management for people with health complexity is increasingly recommended in primary health care (PHC). The INTERMED complexity assessment grid showed adequate psychometric properties in specialized settings. This study aimed to evaluate INTERMED's validity and feasibility to assess health complexity in an adult PHC population.

### Method

The biopsychosocial health care needs of 230 consecutive adult patients from three Brazilian PHC services were assessed using the INTERMED interview. Participants with a total score >20 were classified as "complex". Quality of life was measured using the World Health Organization Quality of Life BREF (WHOQOL-BREF); symptoms of anxiety and depression using the Hospital Anxiety and Depression Scale (HADS); social support using the Medical Outcomes Study—Social Support Survey (MOS-SSS); comorbidity levels using the Charlson Comorbidity Index (CCI). We developed two questionnaires to evaluate health services use, and patient perceived feasibility of INTERMED.

### Results

42 participants (18.3%) were classified as "complex". A moderate correlation was found between the total INTERMED score and the total scores of WHOQOL-BREF (rho = - 0.59) and HADS (rho = 0.56), and between the social domains of INTERMED and MOS-SSS (rho

**Funding:** CAO This study was financed in part by the "Coordination of Superior Level Staff Improvement – Brazil (CAPES) – Finance Code 001". https://www.capes.gov.br/. It was also funded by the "Foundation for Support to Teaching, Research and Assistance at Clinics Hospital of Ribeirão Preto Medical School of University of São Paulo – Brazil (FAEPA)" - https://www.faepa.br/ The funders had no role in study design, data collection and analysis, decision to publish, or preparation of the manuscript.

**Competing interests:** The authors have declared that no competing interests exist.

= -0.44). After adjustment, the use of PHC ($\beta$ = 2.12, t = 2.10, p < 0.05), any other health care services ($\beta$ = 3.05, t = 3.97, p < 0.01), and any medication ($\beta$ = 3.64, t = 4.16, p < 0.01) were associated with higher INTERMED scores. The INTERMED internal consistency was good ($\omega$ = 0.83), and the median application time was 7 min. Patients reported satisfaction with the questions, answers, and application time.

## Conclusion

INTERMED displayed good psychometric values in a PHC population and proved promising for practical use in PHC.

## Introduction

"Health complexity" can be defined as *"interference with the achievement of expected or desired health and cost outcomes due to the interaction of biological, psychological, social, and health systems factors"* [1]. These factors interact dynamically and non-linearly in an idiosyncratic manner for each individual [2–4]. The importance of incorporating the assessment of health complexity into the management of PHC patients for delivering high-quality care with desirable outcomes is increasingly recognized worldwide [5–7]. This is particularly critical in PHC settings that include one of the most medically and socially vulnerable groups of patients in the world, such as Brazil [5]. Despite this acknowledgement, healthcare complexity assessment has been suboptimal or has not been integrated into PHC practice at all, partially because efficient methods for its evaluation in adult PHC are still lacking [8].

The INTERMED Complexity Assessment Grid (adult version) is an efficient tool for assessing biopsychosocial complexity and improving the communication flow between professionals, patients, and services [9]. According to a systematic review, it is one of the best instruments of its kind [10]. Its development was methodologically robust [11], with validation in diverse populations (in- and outpatients in secondary, tertiary, and emergency services, with a range of health problems [12, 13]), using different versions (face-to-face interview [14], self-assessment [15], pediatric [16], adults [14], and elderly [13]), and showing predictive validity regarding mortality [17], healthcare costs [18], and quality of life [19]. Case managers already use INTERMED to identify and coordinate comprehensive care for people with high levels of health complexity [20]. However, to date, only one small study (n = 55) assessed the psychometric properties of INTERMED in a PHC context, which focused mainly on the self-assessment version, and of which no full-text article is available [8].

This study aimed to evaluate the psychometric properties of the INTERMED Complexity Assessment Grid Adult Interview version in a Brazilian PHC population. We hypothesized that INTERMED could have adequate validity and feasibility (applicability and acceptability) in a PHC population.

## Methods

### Sampling technique

We aimed for a 10:1 ratio (i.e., 10 patients per each of the 20 INTERMED items) [21, 22], resulting in an estimated minimum of 200 patients and their health care records. Assuming a dropout rate of 15% in a planned second wave of the collection (not reported here), we decided to recruit an additional 30 people. To maximize the chance of gathering representative data,

we deployed a quota sampling method by dividing the population into gender and age groups (18–30, 31–40, 41–50, 51–60, and ≥60 years) [23].

## Recruitment procedures

Patients were recruited from three PHC services located in Ribeirão Preto city (São Paulo, Southeast region of Brazil).

All adults (age ≥ 18 years) who sequentially arrived at the reception of one of the health services, whether for medical appointments, to pick up medication at the service's pharmacy, to accompany another patient, or to schedule appointments, were approached by a researcher. To be recruited, patients were required to live in the area covered by the service, have their health records there, and speak, read, and write Portuguese at a sufficient level to complete the instruments. Patients who were unable to understand the interview (e.g., due to cognitive impairment or learning disability) or did not complete all the instruments were excluded. When a patient chose not to participate in the study, the researcher invited the next sequential patient. We obtained informed written consent before participation, including consent to review the health care records.

## Instruments

**Participant demographic characteristic questions.** We collected five types of demographic information from the participants: gender, age, ethnicity, education, and occupation.

**INTERMED.** The INTERMED tool is a semi-structured interview that synthesizes data from four health-related aspects (domains): 1) biological, 2) psychological, 3) social, and 4) health system, assessed in the context of time (history, current state, and vulnerability/prognosis) [14, 24, 25]. Within each of the four domains, there are five specific variables (items), totaling 20. Each item has specific clinical anchor points described, ranging from 0 (no vulnerability/only health education) to 3 (severe vulnerability/need for immediate or intensive care) [24] (Fig 1). In previous studies carried out in specialized services, a 20/21 cutoff was proposed to differentiate "complex" from "non-complex" cases [26].

The INTERMED interview involves 16 lead questions related to the four domains and one question about satisfaction with the interview. Based on the information obtained from the

| Domain | History | Current State | Vulnerability |
|---|---|---|---|
| Biological | Chronicity (0) (1) (2) (3) Diagnosis Dilemma (0) (1) (2) (3) | Symptom Severity (0) (1) (2) (3) Diagnostic Challenge (0) (1) (2) (3) | Complications and Life Threat (0) (1) (2) (3) |
| Psychological | Restriction in Coping (0) (1) (2) (3) Psychiatric Dysfunction (0) (1) (2) (3) | Resistance to Treatment (0) (1) (2) (3) Psychiatric Symptoms (0) (1) (2) (3) | Mental Health Threat (0) (1) (2) (3) |
| Social | Job and Leisure Problems (0) (1) (2) (3) Social Dysfunction (0) (1) (2) (3) | Residential Instability (0) (1) (2) (3) Poor Social Support (0) (1) (2) (3) | Social Vulnerability (0) (1) (2) (3) |
| Health Sustem | Access to Care (0) (1) (2) (3) Treatment Experience (0) (1) (2) (3) | Organization of Care (0) (1) (2) (3) Coordination of Care (0) (1) (2) (3) | Health System Impediments (0) (1) (2) (3) |

(0) No vulnerability/Only health education     (1) Mild vulnerability/need for monitoring or prevention

(2) Moderate vulnerability/need for treatment or inclusion in treatment     (3) Severe vulnerability/need for immediate or intensive care

**Fig 1. INTERMED domains and temporal context with their variables.**

answers to the 16 questions mentioned, the health professional assesses the four items of vulnerability/prognosis (one for each domain). A health professional is free to follow the topic guide, skip, or modify specific questions according to what a patient said spontaneously. The Portuguese Brazilian version of these questions was used, which underwent cultural adaptation and proved valid for use in an inpatient population in Brazil [27]. Two researchers responsible for data collection and/or health record review, an occupational therapist (CAO), and a nurse (LLC) were trained in using the INTERMED by the authors of its Brazilian Portuguese version [27].

**Hospital Anxiety and Depression Scale (HADS).**  The HADS is a 14-item scale designed to assess anxiety (HADS-A) and depression (HADS-D) symptoms in medical patients [28]. Items are rated on a 4-point severity scale (0 to 3), where the higher score indicates a worse condition. The total score is the sum of the 14 items, with 7 items per subscale. For each subscale, the score is the sum of the seven items (ranging from 0 to 21). The HADS was used in previous research to assess the validity of the INTERMED psychological domain [11].

**Medical Outcomes Study–Social Support Survey (MOS-SSS).**  The MOS-SSS is a 19-item scale designed to assess social support in medical patients [29]. The 19 items cover four domains (emotional/informational support, instrumental support, positive social interaction, and affection). Items are rated on a 7-point Likert rating scale (1–7). The overall score is the mean score of all items and ranges from 1 to 7; a higher score indicates a higher level of perceived social support.

**WHO Quality of Life–Bref (WHOQOL-BREF).**  The WHOQOL-BREF is a 26-item scale consisting of four domains: physical health (7 items), psychological health (6 items), social relationships (3 items), and environmental health (8 items), as well as QOL and general health items [30]. Items are rated on a 5-point rating scale (1–5), which is stipulated as a five-point ordinal scale. The scores are then transformed linearly to a scale of 0–100.

**Charlson Comorbidity Index (CCI).**  The CCI evaluates the comorbidity level. It consists of 19 selected conditions, including 18 physical health conditions and dementia, which are weighted from 1 to 6 and summed to an index on a 0–33 scale [31]. A higher score reflects the greater number and the seriousness of comorbid diseases.

**Questions to assess health system use.**  To assess health system use, we developed a questionnaire for the evaluation of health services use, with six dichotomous questions based on the SABE study [32]. The questions explored three aspects of service use within a six-month period (prior to the interview). Divided in three domains: 1) PHC use ("Have you consulted with a PHC professional in the last six months, excluding today?"); 2) other health care services (than PHC) ("Have you been admitted to a hospital?"; "Have you consulted with a specialist physician?"; "Have you had a consultation at a specialized mental health service?"; "Have you had a consultation at an A&E department?") and; 3) use of any medication ("Do you take any medications?"). In the PHC use variable, patients scored with "1" if they had been in an appointment within six months before the survey, excluding the day of the interview. For the variable of other health care services (than PHC), which had multiple questions, only one "yes" answers were counted, regardless of the corresponding questions. This is because the aim was to know whether the patient had used such services, and not the quantity or type of services used.

**Feasibility questionnaire.**  To examine the patient-perceived feasibility of INTERMED use, we developed a questionnaire with seven questions, each with five Likert response options [33], divided posteriorly into dichotomous groups (satisfactory and unsatisfactory in relation to feasibility). The questions focused on the acceptability of INTERMED (the understanding of each question, how to answer the question, and the length of the interview), and

applicability (the relevance of asking the questions within each of the four domains). The feasibility questionnaire was administered shortly after the patient completed the INTERMED.

## Data collection procedures

The study was conducted between November 2018 and June 2019. To determine the order of data collection across the three PHC units, we followed the daily routines of those units for a week. After this, data collection took place in each PHC unit for two months, from Monday to Friday, between 7 a.m. and 5 p.m. One researcher (CAO) administered all the listed instruments, first the INTERMED, then the questionnaire about the feasibility and, finally, the other instruments (HADS, MOS-SSS, WHOQOL-BREF, and CCI). For each participant, the CAO measured the time taken to apply INTERMED. All data were collected and managed using Research Electronic Data Capture (REDCap) [34], a web-based platform data capture tool hosted at the Ribeirão Preto Medical School of University of São Paulo (Department of Social Medicine) - https://research.fmrp.usp.br/.

## Health records review

Following primary data collection using the listed instruments, participants had their health records (both paper and electronic) jointly reviewed by two researchers (CAO, LLC) using their clinical and INTERMED knowledge. The purpose of analyzing the health records was to understand if PHC health professionals can obtain biopsychosocial information from existing patient data, without a need for conducting INTERMED interviews (*i.e.*, as a mean to evaluate the practical applicability of the tool evaluate, as an aspect of feasibility). INTERMED questions were applied to health records, and any information that could be filled out completely on the instrument was marked as present. The health records data did not influence the INTERMED interview scores. In previous research, INTERMED was used in a similar way [35].

## Statistical analysis

All data analyses were conducted using the free and open software Jamovi version 1.6.12.

**Descriptive statistics of the sample and INTERMED's feasibility.** The remaining data were summarized using simple descriptive statistics. We calculated frequencies for: 1) demographic characteristics of the study population (i.e., gender, age, ethnicity, education, and occupation categories); 2) patient responses to the seven questions on the acceptability and applicability of INTERMED, and 3) information on each of the INTERMED variables identified in the health records [35].

**Convergent validity.** We performed the Shapiro-Wilk test to examine the distribution of INTERMED data associated with the other four instruments (i.e., HADS, MOSS-SSS, WHOQOL-BREF, and CCI). The results did not meet the prerequisites for normality and homogeneity; therefore, Spearman's correlation analysis was performed. Spearman coefficients (rho) ranging from 0.10 to < 0.40, from 0.40 to < 0.70, from 0.70 to < 1.00, were interpreted as weak, moderate and strong respectively [36].

**Predictive validity.** Hierarchical multiple linear regression analysis was used to test if "health system" use could predict INTERMED-based biopsychosocial complexity. One dependent variable was entered into the model (i.e., continuous INTERMED scores) and the three independent dichotomous variables (i.e., PHC use, other health care services (than PHC), and use of any medication). We conducted a study in three stages to select the best predictive model: first, we built the model through forward selection using the AIC as a criterion and controlling the parameters of age and sex. Second, we analyzed the coefficient determination

ratio to detect the influential points and build a new model without the influential points using the same AIC criterion [37, 38]. Finally, we analyzed the Shapiro-Wilk test to diagnose the model and to confirm that the withdrawal of the influential outliers was consistent with the normality requirement of the regression model.

**Internal consistency.** The internal consistency of INTERMED was measured by omega coefficient and with values ranging from <0.5, from 0.5 to 0.6, from 0.6 to 0.7, from 0.7 to 0.8, from 0.8 to 0.9, and ≥ 0.9, which were interpreted as unacceptable, poor, questionable, acceptable, good, and excellent, as per the interpretation of an alpha coefficient [39].

## Ethics approval

The Research Ethics Committee of the Community Health Center of the Ribeirão Preto Medical School of the University of São Paulo approved the study (n° 99566718.0.0000.5414 in 10/2018).

## Results

We invited two hundred and forty-three (243) patients, of whom five did not agree to participate, and two hundred and thirty-eight (238) agreed. Eight people were excluded because they chose not to complete all the questionnaires. Table 1 shows the socio-demographic characteristics of the 230 participants [mean age = 45.92 (±15.43) years, 56.1% female, 53.5% reported being white, 43.5% reported incomplete higher education or complete high school education, and 40.4% were employed].

The INTERMED minimum total score value was zero, the maximum value was 38, the mean was 13.57 (±7.54), and the median was 13. The INTERMED profiles of the participants according to the clinical anchor points of each item (see S1 Table). A total of 42 (18.3%) participants were classified as "complex", according to the 20/21 cutoff score [21]. Ninety-two

**Table 1. Socio-demographic characteristics of the 230 participants, PHC patients.**

| Characteristic | | Frequency | % |
|---|---|---|---|
| **Age group** | 18–30 | 39 | 17.0 |
| | 31–40 | 59 | 25.7 |
| | 41–50 | 45 | 19.6 |
| | 51–60 | 43 | 18.7 |
| | 60+ | 44 | 19.1 |
| **Gender** | Female | 129 | 56.1 |
| | Male | 101 | 43.9 |
| **Ethnicity** | White | 123 | 53.5 |
| | Black | 23 | 10.0 |
| | Brown | 84 | 36.5 |
| **Education level** | Illiterate/incomplete primary education | 20 | 8.7 |
| | Primary education /Incomplete secondary education | 51 | 22.2 |
| | Secondary education/high school incomplete | 38 | 16.5 |
| | High school/incomplete higher education | 100 | 43.5 |
| | Graduated | 21 | 9.1 |
| **Occupation** | Employee | 93 | 40.4 |
| | Unemployed | 53 | 23.0 |
| | Retired | 50 | 21.7 |
| | Freelance | 30 | 13.0 |
| | Student | 4 | 1.7 |

**Table 2. Spearman's correlation coefficients between INTERMED and other tools.**

|  | Biological | | Psychological | | Social | | Health system | | INTERMED total score | |
|---|---|---|---|---|---|---|---|---|---|---|
|  | **rho** | **p** | **rho** | **p** | **rho** | **p** | **rho** | **p** | **rho** | **p** |
| **HADS:** |  |  |  |  |  |  |  |  |  |  |
| **Total** | 0.31 | <0.01 | **0.59**[a] | <0.01 | **0.41** | <0.01 | 0.26 | <0.01 | **0.56** | <0.01 |
| **Anxiety** | 0.34 | <0.01 | **0.57** | <0.01 | 0.38 | <0.01 | 0.22 | <0.01 | **0.55** | <0.01 |
| **Depression** | 0.21 | <0.01 | **0.50** | <0.01 | 0.36 | <0.01 | 0.26 | <0.01 | **0.46** | <0.01 |
| **MOS—SSS** | -0.12 | <0.05 | -0.35 | <0.01 | **-0.44** | <0.01 | -0.17 | <0.01 | -0.38 | <0.01 |
| **CCI** | 0.09 | 0.08 | -0.02 | 0.18 | -0.01 | 0.40 | -0.12 | 0.20 | 0.08 | 0.75 |
| **WHOQOL-BREF:** |  |  |  |  |  |  |  |  |  |  |
| **Total** | **-0.44** | <0.01 | **-0.57** | <0.01 | -0.36 | <0.01 | -0.26 | <0.01 | **-0.59** | <0.01 |
| **Physical** | **-0.58** | <0.01 | **-0.52** | <0.01 | -0.32 | <0.01 | -0.28 | <0.01 | **-0.63** | <0.01 |
| **Psychological** | -0.23 | <0.01 | **-0.44** | <0.01 | -0.36 | <0.01 | -0.20 | <0.01 | **-0.44** | <0.01 |
| **Social** | -0.20 | <0.01 | **-0.41** | <0.01 | -0.24 | <0.01 | -0.21 | <0.01 | **-0.42** | <0.01 |

HADS = Hospital Anxiety and Depression Scale; MOS-SSS = Medical Outcomes Study–Social Support Survey; CCI = Charlson Comorbidity Index;

WHOQOL-BREF = World Health Organization Quality of Life–BREF.

[a] Values in bold represent moderate Spearman correlation.

patients (40.0%) presented physical/mental multimorbidity, 34 (14.8%) were considered to have health complexity, and 32 (13.9%) had only physical multimorbidity, of whom 6 (2.6%) were "complex". Of the remaining 106 patients without multimorbidity, 2 (0.9%) were considered "complex".

## Validity

With regard to concurrent validity, there were moderate correlations between the total INTERMED score and its psychological domain with HADS (ranging from 0.46 to 0.59, $p < 0.05$), and between the INTERMED social domain and the total HADS score (0.41, $p < 0.05$). There was a moderate inverse correlation between the INTERMED social domain and MOS-SSS (-0.44, $p < 0.05$), the total INTERMED score and its psychological domain with WHOQOL-BRIEF (-0.44, $p < 0.05$), as well as the INTERMED biological domain with the physical domain of WHOQOL (-0.58, $p < 0.05$) and the total WHOQOL score (-0.59, $p < 0,05$) (Table 2).

The omega coefficient was 0.834, suggesting good internal consistency [39–41]. After deleting each of the 20 items from INTERMED, the omega coefficient values ranged from 0.817 to 0.836. Four items showed no decrease in the original omega coefficient value when deleted: "treatment experience" (0.835), "resistance to treatment," "access to care" (both 0.836), and "job and leisure problems" (0.837).

To verify whether the "health system" use (PHC, other health care services (than PHC), and use of any medication) can predict participants' levels of complexity based on the INTERMED criterion, we used hierarchical multiple linear regression. The analysis resulted in model 1 [$F_{(3,23)} = 14.1$, $p < 0.01$, $R^2 = 0.16$, AIC = 1552] and, after controlling the parameters of age and sex, in model 2 [$F_{(8,22)} = 10.2$, $p < 0.01$, $R^2 = 0.24$, AIC = 1529]. The results of the Shapiro-Wilk test were 0.98 ($p < 0.01$) and 0.98 ($p < 0.01$) respectively, indicating that the assumptions of normality were not met and, therefore, models 1 and 2 were inadequate. Next, through graphical analysis, we identified influential outliers. These were patients who, regardless of the complexity level, either used the health system sporadically (only in A&E) or used the system in an exaggerated way (with excessive consultations in A&E, plus 10 consultations in PHC,

**Table 3. Standard multiple linear regression models for INTERMED and health care use.**

**Model 1: Complexity level ($r^2 = 0.16$)**

| Predictor | Coefficient β | IC (95%) | t | P |
|---|---|---|---|---|
| Intercept [a] | 7.50 | 5.25; 9.75 | 6.57 | <0.01 |
| Use of any medication | **4.24** [b] | **2.35; 6.13** | **4.42** | **<0.01** |
| Use of PHC | **2.72** | **0.41; 5.03** | **2.32** | **0.02** |
| other health care services (than PHC) | **2.74** | **0.92; 4.56** | **2.97** | **0.01** |

**Model 2 after adjustment of confounders: Complexity level ($r^2 = 0.24$)**

| Predictor | Coefficient β | IC (95%) | t | P |
|---|---|---|---|---|
| Intercept [a] | 9.68 | 6.34; 13.01 | 5.72 | <0.01 |
| Use of any medication | **3.57** | **1.61; 5.53** | **3.58** | **<0.01** |
| Use of PHC | 1.55 | -0.67; 3.78 | 1.38 | 0.17 |
| other health care services (than PHC) | **2.79** | **1.06; 4.52** | **3.18** | **0.01** |
| Age groups: | | | | |
| 18–30 vs. >60 | -0.44 | -3.50; 2.61 | -0.28 | 0.77 |
| 31–40 vs. >60 | 0.37 | -2.32; 3.06 | 0.27 | 0.78 |
| 41–50 vs. >60 | 2.64 | -0.13; 5.41 | 1.88 | 0.06 |
| 51–60 vs. >60 | **2.98** | **0.20; 5.76** | **2.11** | **<0.05** |
| Sex | | | | |
| Male vs. Female | **-4.48** | **-6.24; -2.73** | **5.03** | **<0.01** |

**Model 3 after adjustment of confounders and withdrawal of residual outliers: Complexity level ($r^2 = 0.39$)**

| Predictor | Coefficient β | IC (95%) | t | P |
|---|---|---|---|---|
| Intercept [a] | **3.08** | **0.32; 5.85** | **2.20** | **<0.05** |
| Use of any medication | **3.65** | **1.92; 5.38** | **4.16** | **<0.01** |
| Use of PHC | **2.12** | **0.13; 4.11** | **2.10** | **<0.05** |
| other health care services (than PHC) | **3.05** | **1.54; 4.57** | **3.97** | **<0.01** |
| Age groups: | | | | |
| 18–30 vs. >60 | -0.40 | -3.08; 2.27 | -0.30 | 0.77 |
| 31–40 vs. >60 | 0.95 | -1.84; 2.86 | 0.43 | 0.67 |
| 41–50 vs. >60 | 0.15 | -0.30; 4.60 | 1.73 | 0.08 |
| 51–60 vs. >60 | **3.49** | **0.49; 5.35** | **2.37** | **<0.05** |
| Sex | | | | |
| Male vs. Female | **5.74** | **4.20; 7.28** | **7.35** | **<0.01** |

[a] Represents reference level.

[b] Values in bold represent the highest predictor value.

and more than 10 medications). After analyzing and excluding these influential outliers, Model 3 was run [$F_{(8,21)} = 16.8$, $p < 0.05$, $R2 = 0.39$, AIC = 1380], for which the results of the Shapiro-Wilk test were 0.99, $p = 0.77$, indicating that the normality assumption was met and model 3 was adequate. Overall, after adjusting for confounders and excluding the 12 influential outliers (model 3), we found that the use of PHC, other health care services (than PHC), and the use of any medication were predictors of complexity according to the INTERMED criteria. Table 3 presents the model development.

## Feasibility

All patients reported satisfaction with the questions asked, the answers, and the application time. The range of the application time was 3–32 minutes; the average application time was 8.15 minutes; the median was 7 minutes and up to 14 and 18.15 minutes for 90% and 95% of

the patients respectively. The perceived relevance of the domains was as follows: biological (n = 230, 100%), psychological (n = 227, 98.7%), health system (n = 221, 96.1%), and social (n = 215, 93.5%). The health records analysis showed that the psychological, social, and health system domains had incomplete data (S2 Table). Only the INTERMED biological domain had more than 50% of the items already described in the health records.

## Discussion

We explored the validity and feasibility of the INTERMED adult interview tool applied in PHC attendees in Brazil, using an adequate sample size, multiple performance metrics, and exploring patients' opinions. To the best of our knowledge, this is the second in general, and the first as thorough and fully reported assessment of INTERMED in a PHC population. We found moderate Spearman's correlation coefficients between the four INTERMED domains and other instruments based on a comprehensive approach to health status (HADS, MOS-SSS, and WHOQOL-BREF). Similar results have been reported for INTERMED studies in specialized services (ranging from 0.55 to 0.74) [11, 12]. The correlation with the CCI, which only quantifies health conditions [31], was found to be weak, which reflects the completeness of the INTERMED tool. The good results regarding INTERMED internal consistency are similar to those found in other studies [10]. Using multiple linear regression, we found an association between higher INTERMED scores and higher use of PHC, other health services (than PHC), and use of any medication. These results suggest that the INTERMED tool is valid for use in a PHC setting [33, 42].

Previous research has suggested that health records with biopsychosocial data facilitate evidence-based care planning development, with increased communication between patients and health systems [43–45]. We found that the data already existing in the health records were focused almost exclusively on the biological aspect, which is considered a widely reported problem and an area for improvement [46, 47]. These results mean that INTERMED could help assess, organize, and coordinate all relevant biopsychosocial aspects of the health service network's information-sharing process [14].

The maximum of 14 minutes needed to complete the interview for 208 (90%) of the participants was shorter than the recommended duration of a single outpatient appointment in Brazil [48], and the median of 7 minutes is compatible with the average PHC appointment duration in 39 countries [49]. The median time being relatively shorter is likely to be related to the fact that 127 (55%) of the interviewed patients were classified as "non-complex" and did not require further clarification after applying INTERMED [49]. This is different from what was found in specialized services, in which the INTERMED application time ranged from 20 to 40 minutes, with a smaller percentage of patients being considered "non-complex" [27]. Another previous study, in a PHC context, also supports the position that "non-complex" consultations are significantly shorter than "complex" consultations [50]. These results regarding the application time were obtained by applicators with the theoretical and practical training proposed by the authors of the INTERMED [25]. The application time results, together with patient perceived feasibility, suggest that INTERMED is a promising candidate for practical use in the Brazilian PHC context [48].

## Study limitations

While we utilized the cutoff point applied in all previous studies, the clinical significance of these cutoff points in PHC is unclear. The cutoff in the context of PHC could be established through the application of ROC analysis, by measuring the sensitivity and specificity of different cutoff scores and their relationship to variables found in a larger sample of PHC patients.

## Research implications

Given the appropriate psychometric properties of INTERMED in the referred sample, in future research, the authors plan to: 1) evaluate its implementation in routine PHC practice to assist person-centered care planning for better health outcomes; 2) evaluate its implementation in primary and specialized services within the same health service network to enable integrated care for better health and service outcomes; 3) develop digital versions of INTERMED to enable objectives 1 and 2; 4) assess whether the future use of health services can be predicted from an INTERMED score; and 5) evaluate INTERMED psychometric properties in other PHC populations and contexts.

## Conclusions

This study showed that INTERMED has adequate psychometric properties to help PHC teams assess the biopsychosocial complexity of health needs. INTERMED could assist PHC professionals and teams in defining patient complexity profiles and developing healthcare planning. The results indicate the need for further studies to assess the potential of INTERMED to enable the delivery of integrated and person-centered care.

## Supporting information

**S1 Table. Profiles of the 230 PHC patients regarding INTERMED items and their clinical anchor points.**
(DOCX)

**S2 Table. Completeness of the INTERMED's domains in the health records.**
(DOCX)

## Acknowledgments

We would like to thank Professor Craig Ramsay for his comments on this work during an international meeting held at the Health Services Research Unit of the University of Aberdeen in July 2019.

## Author Contributions

**Conceptualization:** Camila Almeida de Oliveira, Magdalena Rzewuska, João Mazzoncini de Azevedo-Marques.

**Data curation:** Camila Almeida de Oliveira.

**Formal analysis:** Camila Almeida de Oliveira, Jair Lício Ferreira dos Santos, Miriane Lucindo Zucoloto, Lisa Laredo de Camargo, Ana Carolina Guidorizzi Zanetti, Magdalena Rzewuska, João Mazzoncini de Azevedo-Marques.

**Investigation:** Camila Almeida de Oliveira, Lisa Laredo de Camargo.

**Methodology:** Camila Almeida de Oliveira, Ana Carolina Guidorizzi Zanetti, Magdalena Rzewuska, João Mazzoncini de Azevedo-Marques.

**Supervision:** Ana Carolina Guidorizzi Zanetti, Magdalena Rzewuska, João Mazzoncini de Azevedo-Marques.

**Writing – original draft:** Camila Almeida de Oliveira, Magdalena Rzewuska, João Mazzoncini de Azevedo-Marques.

**Writing – review & editing:** Camila Almeida de Oliveira, Bernardete Weber, Jair Lício Ferreira dos Santos, Miriane Lucindo Zucoloto, Lisa Laredo de Camargo, Ana Carolina Guidorizzi Zanetti, Magdalena Rzewuska, João Mazzoncini de Azevedo-Marques.

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
