## [Decision Letter · Decision Letter 0]

8 Feb 2021

PONE-D-20-31324

Health Complexity Assessment in Primary Care: a validity and feasibility study of the INTERMED tool.

PLOS ONE

Dear Dr. Oliveira,

Thank you for submitting your manuscript to PLOS ONE. After careful consideration, we feel that it has merit but does not fully meet PLOS ONE’s publication criteria as it currently stands. Therefore, we invite you to submit a revised version of the manuscript that addresses the points raised during the review process.

We look forward to receiving your revised manuscript.

Kind regards,

Stefan Hoefer

Academic Editor

PLOS ONE

Additional Editor Comments (if provided):

Based on the reviewers comments, I would like you to focus in particular on presenting your study more concisely, and the statistical analysis and results must be presented more clearly. Please try to address all the comments of the reviewers very carefully. 

Journal Requirements:

2) In your Methods section, please provide a justification for the sample size used in your study, including any relevant power calculations (if applicable).

3) We suggest you thoroughly copyedit your manuscript for language usage, spelling, and grammar. If you do not know anyone who can help you do this, you may wish to consider employing a professional scientific editing service.  

4)  We note that you have indicated that data from this study are available upon request. PLOS only allows data to be available upon request if there are legal or ethical restrictions on sharing data publicly. For information on unacceptable data access restrictions, please see http://journals.plos.org/plosone/s/data-availability#loc-unacceptable-data-access-restrictions.

5)  Thank you for stating the following in the Acknowledgments/Funding Section of your manuscript:

[This study was financed in part by the “Coordination of Superior Level Staff Improvement

– Brazil (CAPES) – Finance Code 001”. It was also funded by the “Foundation for Support

to Teaching, Research and Assistance at Clinics Hospital of Ribeirão Preto Medical School

of University of São Paulo – Brazil (FAEPA)”. A discussion regarding it occurred during an

international meeting funded by the “Global Challenge Research Fund (GCRF) - Internal

Pump Priming Fund Round 5 of the University of Aberdeen”. The funding sources had no

role in the design, conduct, and reporting of the study.]

 [CAO

This study was financed in part by the “Coordination of Superior Level Staff

Improvement – Brazil (CAPES) – Finance Code 001”. https://www.capes.gov.br/.

It was also funded by the “Foundation for Support to Teaching, Research and

Assistance at Clinics Hospital of Ribeirão Preto Medical School of University of São

Paulo – Brazil (FAEPA)” - https://www.faepa.br/

The funders had no role in study design, data collection and analysis, decision to

publish, or preparation of the manuscript.]

Reviewers' comments:

Reviewer's Responses to Questions

**Comments to the Author**

1. Is the manuscript technically sound, and do the data support the conclusions?

Reviewer #1: Yes

Reviewer #2: Partly

2. Has the statistical analysis been performed appropriately and rigorously? 

Reviewer #1: No

Reviewer #2: No

3. Have the authors made all data underlying the findings in their manuscript fully available?

Reviewer #1: No

Reviewer #2: No

4. Is the manuscript presented in an intelligible fashion and written in standard English?

Reviewer #1: No

Reviewer #2: Yes

5. Review Comments to the Author

Reviewer #1: This study aimed to evaluate the feasibility and validity of the application of the INTERMED interview in primary health care.

The INTERMED is an interview to assess the bio-psycho-social health care needs of patients. An advantage of the INTERMED is that many variables assessed in the interview are directly related to health care decisions. Using a cut-off 20/21 one can identify complex patients in need of integrated care. In general practice – with a high time pressure - the application of the INTERMED before a consultation could be of high importance to detect complexity or bio-psycho-social problem areas that are not detected by the standard consultation. The present study is therefore important and useful. In addition, with n=230 interviewed patients in primary health care the sample appears large enough to evaluate feasibility and validity.

However, I have several concerns regarding the manuscript (abstract, introduction, methods etc.). I therefore recommend a major revision.

In addition, the manuscript has to be revised by a native speaker.

More detailed comments:

Abstract:

• The abstract is hard to understand due to language problems

• The INTERMED has to be shortly explained – was it the interview or the questionnaire? What does it measure?

• The other assessment instruments have to be listed (methods).

• How is “health complexity” defined?

• Please shorten the description of the use of specific correlation coefficients in the abstract

• “Spearman’s correlations located between 0.44 – 0.65”. Which correlations are reported? The INTERMED total score correlated with which variables to which amount? This must be described exactly and clearly. The same holds for the other correlations (or just name a few variables and their correlations – but use exact descriptions).

• 7 minutes for an INTERMED interview is a very short time. How did this happen? (To be explained in the results section or discussion)

Introduction:

• Please ask a native speaker to revise the manuscript

• Please insert line numbers in the revision

• “identifying the epidemiology of patient complexity” – what is meant by this? The reference cited (9) does not refer to an epidemiological study.

• Please reduce the number of references considerably

• “how to effect the identification of complex patients” – I do not understand this sentence.

• The introduction must be re-written. It is not clear what the study wants to show. Perhaps the authors want to say that complexity is frequent in general practices but not so easy to detect? And that the INTERMED interview could be a useful assessment instrument for a GP. However, to date, only one study evaluated the usefulness and validity of the INTERMED interview in the frame of GPs. And, therefore, the present study aims to …

Methods:

• Was the (already recruited) sample divided into groups or the sampling based on various age groups?

• Instruments: The description of the INTERMED is hard to understand. Perhaps the authors could insert a Figure showing the INTERMED grid. The INTERMED interview assesses four domains: biological, psychological, social, and health care use. In each domain, five variables are rated on a scale ranging from 0-3, resulting in 20 scores. Each score ranges from zero evidence for or health service need (0), to a clear and serious disturbance or health service needs (3).

• There are 16 lead questions for the interview. The remaining 4 items are prognostic items – that is, the interviewer gives a prognostic score (without a lead question).

• Why was Spearman’s correlation coefficient used? Why not Pearson for the correlation between INTERMED total score and HADS score, for instance?

• What is meant by Pearson X2? Is it a chi-squared test? What is compared here? A chi-squared test compares groups with dichotomous outcomes. Were Pearson correlation coefficients calculated? Please clarify. The distribution of health care use is probably skewed. Here, a Pearson correlation is probably not the best choice.

Results:

• How many patients refused to participate? What are the differences between non-responders and responders? What were the reasons to refuse?

• Validity: The correlations ranged from 0.44 to 0.65? In Table 3, there are smaller correlations presented.

• I do not understand Table 4. In Table 4, chi-squared statistics and p-values are presented. What was calculated here? Which groups were compared? Were Pearson correlations calculated? Then the correlation coefficients should be shown, together with a p-value.

• Please shorten the paragraph regarding “feasibility”.

Discussion:

• The discussion must be shortened. Some parts are hard to understand due to language

Reviewer #2: !. Is the Pearson chi square test appropriate when one variable is a parametric test score (i.e., scores on the INTERMED current status)?

2. Why was Spearman rho used as the correlation coefficient? Were the data in question rank ordered?

3. Is coefficient alpha appropriate for an instrument that is not tau equivalent? That is, can it be demonstrated that the INTERMED is an unidimensional instrument appropriate for the alpha coefficient?

4. Why was the term "construct validity" used on page 9?

5. What is an "academic clinician" (page 9)?

6. On page 7 is the following statement: "we evaluated the concurrent validity of each of its four domains with other well-validated specific instruments for these domains." The next sentence starts listing the other instruments, the first of which is the Socio-demographic Questionnaire that was "developed for the study." Is the assertion that this instrument was "well-validated?" Or, the adaptation resulting in the Questionnaire for Evaluation of Health Services Use? These scales were not specifically used in the computation of the validity coefficients, but nevertheless there does seem to be a gap between the Term "well-validated" and documentation thereof, including documentation of the validity for criterion variables HADS, MOS-SSS, CCI, and WHOQOL-BREF.

7. Was consideration given to the control of type 1 error in the series chi square tests, and Spearman correlations? If not, why not?

6. PLOS authors have the option to publish the peer review history of their article (what does this mean?). If published, this will include your full peer review and any attached files.

Reviewer #1: No

Reviewer #2: No

---

## [Author Response · Author response to Decision Letter 0]

25 May 2021

We are grateful for all the comments. Our point by point response to each comment can be found in a separate document entitled 'response to reviewers

---

## [Decision Letter · Decision Letter 1]

26 Jul 2021

PONE-D-20-31324R1

Health Complexity Assessment in Primary Care: a validity and feasibility study of the INTERMED tool.

PLOS ONE

Dear Dr. Oliveira,

Thank you for submitting your manuscript to PLOS ONE. After careful consideration, we feel that it has merit but does not fully meet PLOS ONE’s publication criteria as it currently stands. Therefore, we invite you to submit a revised version of the manuscript that addresses the points raised during the review process.

We look forward to receiving your revised manuscript.

Kind regards,

Stefan Hoefer

Academic Editor

PLOS ONE

Additional Editor Comments (if provided):

Both reviewers belief that your manuscript is of importance and merits publication in PLOS ONE. I kindly ask you to in particular focus on the comments of reviewer 1 and try to thoroughly address the raised statistical issues as well as the language concerns.

Reviewers' comments:

Reviewer's Responses to Questions

**Comments to the Author**

1. If the authors have adequately addressed your comments raised in a previous round of review and you feel that this manuscript is now acceptable for publication, you may indicate that here to bypass the “Comments to the Author” section, enter your conflict of interest statement in the “Confidential to Editor” section, and submit your "Accept" recommendation.

Reviewer #1: (No Response)

Reviewer #2: All comments have been addressed

2. Is the manuscript technically sound, and do the data support the conclusions?

Reviewer #1: Partly

Reviewer #2: Yes

3. Has the statistical analysis been performed appropriately and rigorously? 

Reviewer #1: No

Reviewer #2: (No Response)

4. Have the authors made all data underlying the findings in their manuscript fully available?

Reviewer #1: No

Reviewer #2: Yes

5. Is the manuscript presented in an intelligible fashion and written in standard English?

Reviewer #1: No

Reviewer #2: Yes

6. Review Comments to the Author

Reviewer #1: I still believe that the study is worth publishing. The manuscript hast been revised and has improved in quality and formal standards. However, it still lacks scientific precision and good linguistic expression. I would therefore recommend a comprehensive second revision based on the following comments:

Abstract:

The manuscript has been revised and the language has improved. However, I still have the impression that the manuscript was not edited by a native speaker (e.g. expressions such as “18.3 % of the patients were classed as complex” in the abstract or “This study aimed to assess… to assess…). Also, for instance, line 232: “Patients were excluded if they were determined to be unable to consent. In my opinion, the language of the manuscript must be improved.

• In the Methods Section of the Abstract the cut-off for complexity must be mentioned.

• The Methods Section of the Abstract must be re-written. Example for a better formulation: Biopsychosocial health care needs were assessed by using the INTERMED interview… Participants with a total INTERMED score >20 were classified as complex / having complex health care needs. Quality of life was measured by …

• The Results Section of the Abstract should be re-phrased. Example: Results showed a high correlation between INTERMED total score (not “whole” score) and quality of life (r=0.59).

• The above formulations are just examples. The next paragraphs also lack scientific precision. I would really ask the authors to scan other scientific papers and use appropriate formulations. The whole study decreases in value if the expressions are not precise and adequate.

• T- and beta-values should be rounded to two decimal points (instead of three).

• the sentence “In a multiple linear regression analysis, the use of PHC (β = 2.121, t = 2.100, p=0.037), services other than PHC (β = 3.052, t = 3.970, p<0.001) and medication (β = 3.654, t = 4.164, p<0.001) predicted the INTERMED score.” is hard to understand. Perhaps a change to “ the use of PHC (), any other health care service (than PHC) () and the use of any medication () were associated with a higher INTERMED score.

• I do not understand the variable “PHC use”. All the patients were recruited in a primary health care setting. So, everybody score a 1 on this variable?

Introduction:

• The introduction must be revised by a good native speaker

Methods:

• Sampling technique: I do not understand the “rule of thumb” – five observations per variable. The validity of the INTERMED was measured by using the total score (and not by calculating correlations with each item).

• Sampling technique: The authors state that previous NTERMED studies used different sample sizes. These studies varied widely in object and settings. So, why should there be a common sample size calculation for all these studies? An INTERMED study does not need specific assumptions for a sample size calculation (compared to other validation studies). For a validation study one can, for instance, calculate the appropriate sample size based on the assumed correlation between the main instrument (INTERMED) with the gold standard with which I compare the instrument. However, very few validation studies do a power calculation prior to the beginning (see for instance https://hqlo.biomedcentral.com/articles/10.1186/s12955-014-0176-2

• One could also just describe that the sample was collected consecutively over a specific time period – if this was the case – resulting in n=230 patients.

• Line 310: Please change “The instruments were administered between November 2018 to June 2019.” Into “The study was conducted between November 2018 to June 2019” or “Patients were included between November 2018 to June 2019” – or something similar. Please adjust other formulations to a more scientific language.

• Line 293: The patients were recruited in a PHC setting. So, each patient scored a 1 on the variable “PHC use”?

• Line 343: adjusting for sex and gender – or controlling for…

• Lines 344-346: This is not to understand. Outliers are removed beforehand (when data appears to be absolutely implausible). Also, the language is difficult to understand. What is meant by “influent”? How many data was withdrawn from the sample? I would recommend to skip step 2 and 3.

• Line 351: Why was a factor analysis conducted? I would recommend to omit this procedure. It is not part of the study aim. Also, in my opinion, the INTERMED is not uni-dimensional. It consists of four different domains.

• Descriptive statistics should be described first. Followed by the more complex statistics.

Results:

• Table 2 should be omitted or changed in a Supplemental Table

• Please round correlation and p-values to two decimal points.

• Line 497: What means “after controlling for type 1 error?”

• Lines 503-505: The INTERMED is not assumed to be uni-dimensional. Somatic issues (for instance) are not on the same dimension as psychological issues. Perhaps the factor analysis should be omitted. It is also not clear, which factor analysis has been conducted.

• The outliers should not be excluded. These are the patients who use the system excessively (or not at all). There is no reason to exclude them. In contrast, the patients with the highest health care use are probably those with the highest INTERMED scores.

• Table 5 must be omitted.

Discussion:

• line 627 ff: a median does not range. The range of the time was 3-32, the median was 7.

• Lines 637 ff: The cut-off could be investigated with the present study data. One could run an ROC analysis with different cut-offs for complexity and look at the sensitivity and specificity of the various cut-offs regarding a specific outcome (this would need a categorization of the patients according to an external criterion). So, the authors do not have to do this. But a clustering method is not the method to investigate the appropriateness of a cut-off.

Reviewer #2: My main concerns were related to statistical and psychometric issues. The authors provided an evidence foundation for the use of the Spearman correlation and provided the omega coefficient for non-tau equivalence. These and all other concerns were adequately remedied.

7. PLOS authors have the option to publish the peer review history of their article (what does this mean?). If published, this will include your full peer review and any attached files.

Reviewer #1: No

Reviewer #2: No

---

## [Author Response · Author response to Decision Letter 1]

27 Sep 2021

We are grateful for all the comments. Our point by point response to each comment

can be found in a separate document entitled 'response to reviewers

---

## [Decision Letter · Decision Letter 2]

30 Dec 2021

PONE-D-20-31324R2Health Complexity Assessment in Primary Care: a validity and feasibility study of the INTERMED tool.PLOS ONE

Dear Dr. Oliveira,

Thank you for submitting your manuscript to PLOS ONE. After careful consideration, we feel that it has merit but does not fully meet PLOS ONE’s publication criteria as it currently stands. Therefore, we invite you to submit a revised version of the manuscript that addresses the points raised during the review process.

 Reviewer 1 identified one further aspect. I do believe this is an important issue, which can be addressed and resolved.

We look forward to receiving your revised manuscript.

Kind regards,

Stefan Hoefer

Academic Editor

PLOS ONE

Journal Requirements:

Reviewers' comments:

Reviewer's Responses to Questions

**Comments to the Author**

1. If the authors have adequately addressed your comments raised in a previous round of review and you feel that this manuscript is now acceptable for publication, you may indicate that here to bypass the “Comments to the Author” section, enter your conflict of interest statement in the “Confidential to Editor” section, and submit your "Accept" recommendation.

Reviewer #1: All comments have been addressed

Reviewer #2: All comments have been addressed

2. Is the manuscript technically sound, and do the data support the conclusions?

Reviewer #1: Yes

Reviewer #2: Yes

3. Has the statistical analysis been performed appropriately and rigorously? 

Reviewer #1: Yes

Reviewer #2: Yes

4. Have the authors made all data underlying the findings in their manuscript fully available?

Reviewer #1: Yes

Reviewer #2: (No Response)

5. Is the manuscript presented in an intelligible fashion and written in standard English?

Reviewer #1: Yes

Reviewer #2: Yes

6. Review Comments to the Author

Reviewer #1: The manuscript has very much improved, I congratulate on this work.

However, now - as the manuscript reads very fluent - I noticed one aspect that should still be explained (I did not notice it before and apologize for the late comment. However, I believe that this point is important).

It is stated that the health records of the patients were reviewed. However, it is not clear if the INTERMED interview scores were changed based on this reviews? Or how the information gained in these reviews was transferred to the data used in the analysis. Or what exactly was done with the information gained in the health records reviews (in terms of changing interview scores).

Could you please explain this - and if the INTERMED interview scores were changed- provide information about these changes (did the total scores increase after the review, the amount of change (mean values before the reviews, mean values after the reviews etc.)?

This would provide an insight about the information that is missing when applying the INTERMED interview.

With the inclusion of this explanation and additional information I would recommend the publication of the mansucript.

Reviewer #2: All concerns about the statistical and psychometric aspects of the study have been satisfied. Kudos to the authors for the manuscript modifications and improved writing.

7. PLOS authors have the option to publish the peer review history of their article (what does this mean?). If published, this will include your full peer review and any attached files.

Reviewer #1: No

Reviewer #2: No

---

## [Author Response · Author response to Decision Letter 2]

6 Jan 2022

Thank you for the opportunity to revise our manuscript. We would also like to thank the reviewer 1 for their comment. It was very helpful and insightful. Below we describe how it has been addressed in the revised version of the manuscript.

---

## [Editor Report · Decision Letter 3]

26 Jan 2022

Health Complexity Assessment in Primary Care: a validity and feasibility study of the INTERMED tool.

PONE-D-20-31324R3

Dear Dr. Oliveira,

We’re pleased to inform you that your manuscript has been judged scientifically suitable for publication and will be formally accepted for publication once it meets all outstanding technical requirements.

Kind regards,

Chung-Ying Lin

Academic Editor

PLOS ONE

Additional Editor Comments (optional):

The authors have satisfactorily addressed the last comment made by the previous reviewer. I believe that the present manuscript achieves the standard of publication. 
---

## [Editor Report · Acceptance letter]

7 Feb 2022

PONE-D-20-31324R3 

Health Complexity Assessment in Primary Care: a validity and feasibility study of the INTERMED tool 

Dear Dr. Oliveira:

I'm pleased to inform you that your manuscript has been deemed suitable for publication in PLOS ONE. Congratulations! Your manuscript is now with our production department. 

Kind regards, 

on behalf of

Dr. Chung-Ying Lin 

Academic Editor

PLOS ONE